# Joint Trajectory and IRS Phase Shift Optimization for Dual IRS-UAV-Assisted Uplink Data Collection in Wireless Sensor Networks

**DOI:** 10.3390/s25206265

**Published:** 2025-10-10

**Authors:** Heng Zou, Hui Guo

**Affiliations:** School of Physics and Electronic Information, Henan Polytechnic University, Jiaozuo 454003, China; 212311010015@home.hpu.edu.cn

**Keywords:** Intelligent Reflecting Surface, unmanned aerial vehicle, wireless sensor networks, trajectory optimization

## Abstract

**Highlights:**

Based on practical communication scenarios and existing research, this paper innovatively proposed a dual IRS-UAV-assisted uplink model for WSNs. The dual reflection link between the IRSs was utilized to effectively enhance the system’s transmission rate. Additionally, a novel algorithm was designed to jointly optimize the IRS phase shifts and UAV flight trajectory, aiming to maximize the system sum rate. Experimental results demonstrate that the scheme proposed in this paper is significantly superior to the traditional IRS-UAV trajectory design schemes.

**What are the main findings?**
This paper designed a completely new model of dual IRS-UAV-assisted communication.This paper presents a new trajectory optimization algorithm.

**What is the implication of the main finding?**
Compared with the traditional single IRS-UAV-assisted communication model, the newly proposed model can significantly enhance the sum rate of the communication system.The proposed new algorithm can effectively address the non-convexity issue in optimization problems.

**Abstract:**

Intelligent reflecting surface-assisted unmanned aerial vehicles (IRS-UAVs) have been widely applied in various communication scenarios. This paper addressed the uplink communication problem in wireless sensor networks (WSNs) by proposing a novel double IRS-UAVs assisted framework to improve the pairwise sum rate. Specifically, nodes with relatively short signal transmission distances upload signals via a single-reflection link, while nodes with relatively long distances upload signals through a dual-reflection link involving two IRSs. Within each work cycle, the IRS-UAVs followed a fixed service sequence to cyclically assist all sensor node pairs. We designed a joint optimization algorithm that simultaneously optimized the UAV trajectories and IRS phase shifts to maximize the pairwise sum rate while guaranteeing each node’s transmission rate meets a minimum quality of service (QoS) constraint. Specifically, we introduce slack variables to linearize the inherently nonlinear constraints arising from interdependent variables, thereby transforming each subproblem into a more manageable form. These subproblems are then solved iteratively within a coordinated optimization framework: in each iteration, one subproblem is optimized while keeping variables of others fixed, and the solutions are alternately updated to refine the overall performance. The numerical results show that this algorithm can effectively optimize the flight trajectory of the unmanned aircraft and significantly improve the pairwise total rate of the system. Compared with the two traditional schemes, the average optimization rates are 11.91% and 16.36%.

## 1. Introduction

In recent years, with the continuous evolution of 6G networks and the rapid growth in demand for high-performance wireless communication systems, Intelligent Reflecting Surfaces (IRS) have successfully transitioned from the theoretical exploration phase to practical deployment. They have gradually matured in various complex communication scenarios and become a core enabling technology for next-generation wireless communications. From a hardware perspective, an IRS is a planar array composed of reconfigurable passive reflecting elements, whose core function lies in dynamically adjusting the phase, amplitude, and polarization state of incident electromagnetic waves to achieve active reconstruction of the wireless propagation environment and precise optimization of signal transmission paths [1]. It is this unique technical advantage that has driven the widespread application of IRS in diverse fields such as Integrated Sensing and Communication (ISAC) systems, Wireless Sensor Networks (WSNs), and smart city infrastructure [1,2,3,4].

The core competitiveness of IRS is concentrated in two key dimensions: significant cost advantages and highly flexible deployment characteristics. In terms of cost, a standard IRS panel integrating hundreds of reflecting elements has much lower hardware manufacturing and operation and maintenance costs than traditional small-cell base stations. This cost-effectiveness not only reduces the capital investment for large-scale networking but also provides crucial support for its large-scale application in multi-cell and multi-user scenarios [2,5]. In terms of deployment flexibility, IRS does not rely on dedicated infrastructure and can be flexibly integrated into various existing carriers based on actual communication needs; for example, when deployed on the outer surface of high-rise buildings, it can bypass obstacles such as buildings through signal reflection to construct indirect Line-of-Sight (LoS) links for coverage hole areas; if deployed in complex indoor environments (e.g., large shopping malls, underground parking lots), it can effectively expand the signal coverage range to meet the differentiated communication needs of users in different locations [1,3].

However, IRS still has non-negligible limitations in practical applications. The most prominent one is that the physical position of traditional IRS remains fixed after deployment. This static characteristic makes it difficult to adapt to scenarios where user positions change dynamically when users move beyond the fixed coverage range of IRS, the signal optimization capability of IRS will be significantly attenuated, failing to fully exert its role [1,2]. Furthermore, the fixed deployment mode determines that the coverage range of IRS has rigid constraints, which makes it difficult to meet the demand for continuous and stable communication in dynamic scenarios such as disaster rescue (where user positions are scattered and change in real time), industrial IoT (where mobile sensors continuously move), and urban road monitoring (where vehicles move at high speed) [3,5].

To address the above limitations, the technical characteristics of Unmanned Aerial Vehicles (UAVs) provide an ideal solution. UAVs have autonomous flight capabilities and flexible trajectory control advantages, enabling dynamic adjustment of spatial positions according to real-time needs [4,6]. For this reason, some studies have proposed an integrated design that combines UAVs with IRS: by mounting lightweight IRS panels and supporting control circuits on UAV platforms, a “movable intelligent reflecting system” is constructed. This system can rely on the autonomous path planning function of UAVs to track changes in user positions in real time, providing precise signal reflection services for receivers in different scenarios, thereby meeting the real-time communication needs of users in dynamic environments [4,5].

## 2. Related Works

In ref. [7], the authors compared the performance of the relay system in three modes (UAV-only, IRS-only and IRS-UAV) and maximized the system reachable rate by optimizing the number and deployment height of IRS reflectors. The author in ref. [8] also introduced IRS-assisted UAVs (IRS-UAVs) to optimize the NOMA network. By optimizing the phase of IRS, resource allocation and the trajectory of UAVs, the total energy consumption of the system is reduced. In ref. [9], the authors studied the multi-user symbiotic radio (SR) system assisted by IRS-UAV. They developed an iterative algorithm based on block coordinate descent (BCD) technology to meet the minimum communication rate requirements and minimized the maximum bit error rate (BER) among all ground devices (GDs). In ref. [10], the author proposed to improve the network coverage of the IRS-UAV communication system by increasing the number of base stations (BS). In addition, the joint optimization of multiple UAVs can further improve the performance of the system. In ref. [11], the joint IRS selection and beamforming optimization problem had been investigated in multiple IRS-UAV-assisted anti-jamming device-to-device (D2D) networks. Furthermore, the authors in ref. [12] proposed an IRS-assisted multilayer UAV network and analyzed the symbolic error rate and outage probability.

Although many studies have explored the combination of UAVs and IRSs, most of them focus on the optimization problem in the scenario with only single-reflection links. However, as urban density rises, the number of required cells (or users) has significantly increased. The coverage area of an IRS-UAV is difficult to meet the requirements of a large-scale network. Therefore, increasing the number of IRS-UAVs involved in communication and taking advantage of double-reflection links can greatly improve network coverage. In ref. [13], the novel IRS-assisted cognitive UAV systems (CUAVS) were investigated. However, the IRS cannot be moved freely. The authors studied an IRS-UAV wireless-powered mobile edge computing (MEC) system, where multiple IRS-UAVs were considered between Internet of Things (IoT) devices and the BS to improve the computation bits and energy harvesting in [14], but they overlooked the interconnectedness of IRSs.

### Key Contributions

In this paper, we consider a novel IRS-UAVs-assisted uplink of wireless sensor networks (WSNs), where the BS can provide services to sensor nodes in a farther area through a double-reflection link of two IRSs. Compared with the traditional single IRS-assisted UAV system, this system can significantly enhance the network coverage by simultaneously optimizing the flight trajectories of two UAVs and the beamforming of the IRSs. We formulated the corresponding optimization problem, but due to the intercoupling among the optimization variables, it is difficult to solve. We proposed an algorithm that first divides the original problem into multiple sub-problems and introduces slack variables, then solves them iteratively via a joint optimization framework. The simulation results show that, compared with traditional scheme where UAVs fly from a user’s location to another’s, the proposed algorithm can significantly enhance the sum rate of the system at each stage.

## 3. System Model and Problem Formulation

### 3.1. System Model

In this section, we consider the uplink communication scenario of two IRS-UAVs in Figure 1, which involves two coverage holes: one located proximal to the BS and another positioned distally, comprising $K_{1}$ and $K_{2}$ sensor nodes, respectively. There are two IRS-UAVs that perform the communication mission at altitude *H* employing time division multiple access (TDMA). Continuous trajectories are represented with *N* discrete coordinates. For each pair of sensor nodes corresponding to the two covering holes, they are denoted as $u_{k1}$ and $u_{k2}$. Their locations are represented by $u_{k1}$ and $u_{k2}$, respectively. $q_{1,n}$ and $q_{2,n}$ are the positions of the UAVs, which are assumed to be co-located with the IRSs. The discrete trajectory satisfies the following constraints as(1)$$∥q_{m,n}-q_{m,n+1}∥⩽\frac{V_{\max}T}{N},m=1,2,$$
where $V_{\max}$ represents the maximum flight speed of the UAV, T/N represents the time length of each time slot, *n* demotes the index of time slots and *m* demotes the identification number of the UAV.

Due to the obstruction, the sensor nodes and BS cannot communicate directly with each other. We assume both IRS-UAVs operate with identical service cycles of duration *T*, where sensor nodes within the same coverage hole are allocated equal uplink transmission time slots. In time slot *n*, the distances of $BS-I_{1}$, $BS-I_{2}$, $I_{1}$-$u_{k1}$ and $I_{2}$-$s_{k2}$ are denoted by $d_{b,I_{1,n}}=\sqrt{{∥q_{1,n}∥}^{2}+H^{2}}$, $d_{b,I_{2,n}}=\sqrt{{∥q_{2,n}∥}^{2}+H^{2}}$, $d_{I_{1,n}u_{k1}}=\sqrt{{∥q_{1,n}-u_{k1}∥}^{2}+H^{2}}$, $d_{I_{2,n}s_{k2}}=\sqrt{{∥q_{2,n}-s_{k2}∥}^{2}+H^{2}}$, respectively. The IRS-assisted communication channel is composed of LoS and non-line-of-sight (NLoS) components. As the deployment position of the IRS is raised, the probability of LoS will keep increasing. Especially in remote areas, when the altitude exceeds 40 m, the probability of LoS will approach 100% [15]. In this case, we assume that all the channels of the system are approximately equivalent to LoS channels without obstacles. The channel from node *i* to *j*, with $\left\{i,j\right\}\in \left\{\left(\mathrm{BS},I_{1}\right),\;\left(I_{1},u_{k1}\right),\;\left(I_{2},s_{k2}\right)\right\}$ can be modeled in the following general from(2)$$G_{i,j}=g_{i,j}a_{r}\left(\theta_{i,j}^{r},\upsilon_{i,j}^{r},m_{j}\right)a_{t}^{H}\left(\theta_{i,j}^{t},\upsilon_{i,j}^{t},m_{i}\right),$$
where $g_{i,j}=\beta_{0}^{1/2}e^{-j\frac{2\pi }{\lambda }D_{i,j}}/D_{i,j}^{\alpha /2}$ is the complex channel gain of the i→j link with $\beta_{0}$ denoting its channel power gain at the reference distance of 1 m, λ denoting the signal wavelength, $D_{i,j}$ denoting the distance from node *i* to *j*, and α denoting the path-loss exponent. Moreover, $\theta_{i,j}^{r}\left(or\;\upsilon_{i,j}^{r}\right)\in \left[0,\pi \right]$ and $\theta_{i,j}^{t}\left(or\;\upsilon_{i,j}^{t}\right)\in \left[0,\pi \right]$ denote, respectively, the azimuth (or elevation) angle-of-arrival (AoA) at node *j* and angle-of-departure (AoD) at node *i* with respect to the IRS plane, $m_{1}\left(m_{2}\right)$ denotes the number of elements of $I_{1},\left(I_{2}\right)$, $a_{r}\left(\theta_{i,j}^{r},\upsilon_{i,j}^{r},m_{j}\right)$ denotes the received array response from node *i* to *j* and $a_{t}\left(\theta_{i,j}^{t},\upsilon_{i,j}^{t},m_{i}\right)$ denotes the transmitted array response from node *i* to *j*.

The inter-IRS channel between $I_{1}$ and $I_{2}$, is namely(3)$$\begin{array}{c} G_{I_{1},I_{2}}=\\ \underset{f_{1}}{\underset{︸}{\sqrt{g_{I_{1},I_{2}}}a_{r}\left(\theta_{I_{1},I_{2}}^{r},\upsilon_{I_{1},I_{2}}^{r},m_{2}\right)}}\underset{f_{2}^{H}}{\underset{︸}{\sqrt{g_{I_{1},I_{2}}}a_{t}^{H}\left(\theta_{I_{1},I_{2}}^{t},\upsilon_{I_{1},I_{2}}^{t},m_{1}\right)}}. \end{array}$$

In time slot *n*, the effective channels BS-$I_{1}$-$u_{k}$ and BS-$I_{1}$-$I_{2}$-$s_{k}$, denoted by $h_{u_{k1}}$ and $h_{s_{k2}}$, respectively, are given by(4)$$h_{u_{k1}}=g_{I_{1,n},u_{k1}}^{H}\Phi_{1,n}g_{b,I_{1,n}},$$(5)$$h_{s_{k2}}=g_{I_{2,n},s_{k2}}^{H}\Phi_{2,n}G_{I_{1},I_{2}}\Phi_{1,n}g_{b,I_{1,n}}+g_{I_{2,n},s_{k2}}^{H}\Phi_{2,n}g_{b,I_{2,n}},$$
where $g_{I_{1,n},u_{k1}}^{H}\in C^{1\times m_{1}},\;g_{I_{2,n},s_{k2}}^{H}\in C^{1\times m_{2}}$, $g_{b,I_{1,n}}\in C^{m_{1}\times 1}$ and $g_{b,I_{2,n}}\in C^{m_{2}\times 1}$ denotes the channels from $I_{1}$ to $u_{k1}$, $I_{2}$ to $s_{k2}$, BS to $I_{1}$, and BS to $I_{2}$, respectively. $\Phi_{1,n}$ denotes the reflection matrix of $I_{1}$ and the definition of $\Phi_{2,n}$ is similar, which can be given by(6)$$\begin{array}{c} \Phi_{s,n}\in P_{m_{s}}≜\\ \left\{\Phi \left|\begin{array}{c} \left|{\left[\Phi \right]}_{m,m}\right|=1,\\ {\left[\Phi \right]}_{m,p}=0,m,p\in \left\{1,\ldots ,m_{s}\right\},m\neq p \end{array}\right.\right\},\\ s=1,2. \end{array}$$

The corresponding achievable rates from BS to $u_{k1}$ and BS to $s_{k2}$ in bits per second per Hertz (bps/Hz), denoted by $r_{b,u_{k1},n}$ and $r_{b,s_{k2},n}$, respectively, are thus given by(7)$$r_{b,u_{k1},n}=\log_{2}\left(1+\frac{P{\left|h_{u_{k1}}\right|}^{2}}{\sigma^{2}}\right),$$(8)$$r_{b,s_{k2},n}=\log_{2}\left(1+\frac{P{\left|h_{s_{k2}}\right|}^{2}}{\sigma^{2}}\right),$$
where $\sigma^{2}$ denotes the noise power, and *P* represents the transmission power of BS.

In each time slot, only one pair of sensor nodes establishes a communication link with the BS. At the same time, ensure that the transmission rate of each node can meet the minimum quality of service (QoS) standard, which is expressed as(9)$$\forall \left(r_{b,u_{k1},n},r_{b,s_{k2},n}\right)⩾R_{\mathrm{low}},$$
where $R_{\mathrm{low}}$ represents the minimum communication rate requirement for each node. Thus, the sum rate of the sensor nodes in time slot *n* can be expressed as(10)$$R_{n}=r_{b,u_{k1},n}+r_{b,s_{k2},n}.$$

### 3.2. Problem Formulation

Our research aims to maximize the sum rate under the constraints of the phase shifts Φ of IRSs and the trajectories Q of UAVs, in which $\Phi =\left\{\Phi_{1,n},\;\Phi_{2,n}\right\}$, and $Q=\left\{q_{1,n},\;q_{2,n}\right\}$. To simplify the joint trajectory and phase optimization problem of dual IRS-UAVs, we temporarily ignore two practical constraints in the current model: (1) the response time required for IRS phase reconfiguration, and (2) the energy consumption of the IRS-UAV system (including UAV flight energy and IRS control circuit energy).

Thus, this problem is expressed as(11)$$\begin{array}{c} \left(P1\right)\;\max_{\Phi ,Q}R_{n}\\ \;\;\;s.t.\;\left(1\right)\left(6\right)\left(9\right), \end{array}$$
where Equation (Equation 1) represents the maximum flight speed limit of the UAV, Equation (Equation 6) represents the phase constraint of the IRS, and Equation (Equation 9) represents the QoS constraint.

Due to the existence of binary and coupled variables, this problem is difficult to be solved directly. To address (P1), we propose an algorithm to derive the second-best solution.

## 4. Proposed Algorithm

Taking into account the inherent complexity of the coupling variables, we will obtain an effective solution by splitting the original problem into two sub-problems and solving them iteratively. Firstly, given the trajectories of the UAVs, we obtain the optimal phase shift. Then, we optimize the trajectories of the UAVs using the successive convex approximation (SCA) technique. These two sub-problems can be solved separately through iterative algorithms.

### 4.1. Phase Shift Optimization

When operating on sensor node uk1, the optimal phase of $I_{1}$ can be obtained as(12)$${\left[\Phi_{I_{1,n}}\right]}_{m_{1},m_{1}}=e^{-j\left(\arg\left({\left|g_{I_{1,n},u_{k1}}^{H}\right|}_{m_{1}}\right)+\arg\left({\left[g_{b,I_{1,n}}\right]}_{m_{1}}\right)\right)},$$
where $m_{1}=1,\ldots ,\;M_{1}$. At this time, Equation (Equation 7) can be restated as(13)$$r_{b,u_{k1},n}=\log_{2}\left(1+\frac{Pm_{1}^{2}\beta_{0}^{2}}{\sigma^{2}d_{b,I_{1,n}}^{\alpha }d_{I_{1,n},u_{k1}}^{\alpha }}\right).$$

When operating on sensor node $s_{k2}$, referring to [16,17], it can be shown that the main diagonal of the optimal IRSs’ passive beamforming matrix is given by(14)$${\left[{\hat{\Phi }}_{1,n}\right]}_{m_{1},m_{1}}≜e^{-j\left(\arg\left({\left[f_{2}^{H}\right]}_{m_{1}}\right)+\arg\left({\left[g_{B,I_{1,n}}\right]}_{m_{1}}\right)\right)},$$(15)$${\left[{\hat{\Phi }}_{2,n}\right]}_{m_{2},m_{2}}≜e^{-j\left(\arg\left({\left[g_{I_{2,n},s_{k2}}^{H}\right]}_{m_{2}}\right)+\arg\left({\left[f_{1}\right]}_{m_{2}}\right)\right)},$$
where $m_{s}=1,\ldots ,\;M_{s},\;s=1,\;2$. We assume that the time consumed by the phase optimization of IRSs can be neglected. By optimizing Equations (Equation 13) and (Equation 14), the IRS’s beamforming achieves the optimal phase shifts. Then Equation (Equation 8) can be restated as(16)$$\begin{array}{c} r_{b,s_{k2},n}=\log_{2}\left(1\right.\\ \left.+\frac{P}{\sigma^{2}}{\left(\frac{m_{1}m_{2}\beta_{0}^{\frac{3}{2}}}{{\left(d_{b,I_{1,n}}d_{I_{2,n},s_{k2}}D_{I_{1},I_{2}}\right)}^{\frac{\alpha }{2}}}+\frac{m_{2}\beta_{0}}{{\left(d_{b,I_{2,n}}d_{I_{2,n},s_{k2}}\right)}^{\frac{\alpha }{2}}}\right)}^{2}\right). \end{array}$$

### 4.2. Trajectory Optimization

After acquiring the phase shifts of IRSs, the trajectories of the UAVs can be derived by solving the following optimization problem.(17)$$\begin{array}{c} \left(P2.1\right)\;\max_{q_{1,n}}R_{n}\\ \;\;\;s.t.\;\left(1\right),\left(9\right), \end{array}$$(18)$$\begin{array}{c} \left(P2.1\right)\;\max_{q_{2,n}}R_{n}\\ \;\;\;s.t.\;\left(1\right),\left(9\right). \end{array}$$

It is very difficult to solve (P2.1) directly, so we employ slack variables $d_{1k,n}$ to replace $d_{b,I_{1,n}}^{\alpha }d_{I_{1,n},u_{k1}}^{\alpha }$, satisfying(19)$$d_{1k,n}⩾d_{b,I_{1,n}}^{\alpha }d_{I_{1,n},u_{k1}}^{\alpha }.$$

Therefore, we can rewrite Equation (Equation 13) as(20)$$R_{n}=\underset{\left(a\right)}{\underset{︸}{\log_{2}\left(1+\frac{C_{1}}{d_{1k,n}}\right)}},$$
where $C_{1}=\frac{Pm_{1}^{2}\beta_{0}^{2}}{\sigma^{2}}$. The second-order derivative of (a) with respect to $d_{1k,n}$ can be written as $\frac{\left(2d_{1k,n}+C_{1}\right)C_{1}}{{\left(d_{1k,n}^{2}+d_{1k,n}C_{1}\right)}^{2}\ln2}$. It is obvious that the second-order derivative is larger than 0 when $C_{1}>0$ and $d_{1k,n}>0$ are satisfied. We can use SCA technology to obtain the lower limit ${\hat{R}}_{1k,n}^{\mathrm{low}}$ of (a).(21)$$\begin{array}{c} \begin{array}{cc} r_{b,u_{k1},n} & \geq \log_{2}\left(1+\frac{C_{1}}{d_{1k,n}^{\left(l\right)}}\right)\\ \; & \;-\frac{C_{1}}{\ln2\left({\left(d_{1k,n}^{\left(l\right)}\right)}^{2}+d_{1k,n}^{\left(l\right)}C_{1}\right)}\left(d_{1k,n}-d_{1k,n}^{\left(l\right)}\right)\\ \; & ={\hat{R}}_{1k,n}^{\mathrm{low}}. \end{array} \end{array}$$

The right-hand sides of the constraints Equation (Equation 19) are all non-negative and convex (α⩾2). We can obtain the expressions for the approximations $F\left(k,q_{1,n}\right)$ [13], which are denoted by(22)$$\begin{array}{cc} \; & d_{b,I_{1},n}^{\alpha }d_{I_{1,n},u_{k1}}^{\alpha }\\ \; & \leq \frac{1}{2}((d_{b,I_{1},n}^{\alpha }+d_{I_{1,n},u_{k1}}^{\alpha }{)}^{2}-{\left(d_{b,I_{1,n}}^{\left(l\right)}\right)}^{2\alpha }-{\left(d_{I_{1,n},u_{k1}}^{\left(l\right)}\right)}^{2\alpha })\\ \; & \;-2{\left(d_{b,I_{1,n}}^{\left(l\right)}\right)}^{\alpha }{\left(q_{1,n}^{\left(l\right)}\right)}^{T}\left(q_{1,n}-q_{1,n}^{\left(l\right)}\right)\\ \; & \;-2{\left(d_{I_{1,n},u_{k1}}^{\left(l\right)}\right)}^{\alpha }{\left(q_{1,n}^{\left(l\right)}-u_{k1}\right)}^{T}\left(q_{1,n}-q_{1,n}^{\left(l\right)}\right)\\ \; & =F(k,q_{1,n}). \end{array}$$

From these, we can convert (P2.1) into the following questions:(23)$$\begin{array}{c} \left(P3.1\right)\;\max_{q_{1,n}}R_{n}\\ \;\;\;s.t.\;\left(1\right),\left(9\right)\\ \;\;\;F\left(k,q_{1,n}\right)⩽d_{1k,n}. \end{array}$$

For (P2.2), due to $d_{b,I_{1,n}},d_{b,I_{2,n}},d_{I_{2,n},s_{k2}}>0$ we employ variables $x=\ln d_{b,I_{1,n}},y=\ln d_{b,I_{2,n}}$ and $z=\ln d_{I_{2,n},s_{k2}}$ to replace them. Therefore, we can rewrite Equation (Equation 16) as(24)$$r_{b,s_{k2},n}=\log_{2}\left(1+\frac{P}{\sigma^{2}}{\left(\underset{\left(b\right)}{\underset{︸}{C_{2}e^{-\frac{\alpha }{2}\left(x+z\right)}}}+\underset{\left(c\right)}{\underset{︸}{C_{3}e^{-\frac{\alpha }{2}\left(y+z\right)}}}\right)}^{2}\right),$$
where $C_{2}=\frac{m_{1}m_{2}\beta_{0}^{\frac{3}{2}}}{{D_{I_{1},I_{2}}}^{\frac{\alpha }{2}}},C_{3}=m_{2}\beta_{0}$. It can be shown that the second derivatives of (b) and (c) are greater than zero, and Equation (Equation 24) is a convex function. The suboptimal solution of (P2.2) can be obtained through SCA.

### 4.3. Joint Optimization of Trajectory and Phase Shift Algorithm

Based on the solutions to the sub-problems proposed in the above sections, we can obtain the maximum sum rate by iteratively solving these sub-problems. The algorithm for joint trajectory and phase shift design is summarized in Algorithm 1.
**Algorithm 1** Proposed Solution for (P1)**Initialize**: Set $\left\{\Phi_{1,n}^{\varepsilon },\Phi_{2,n}^{\varepsilon },q_{1,n}^{\varepsilon },q_{2,n}^{\varepsilon }\right\}$ and ε=0.**repeat**       Given $\left\{\Phi_{2,n}^{\varepsilon },q_{1,n}^{\varepsilon },q_{2,n}^{\varepsilon }\right\}$, obtain the optimal phase shift $\Phi_{1,n}^{\varepsilon +1}$ based on Equations (Equation 12) and (Equation 14);       Given $\left\{\Phi_{1,n}^{\varepsilon +1},q_{1,n}^{\varepsilon },q_{2,n}^{\varepsilon }\right\}$, obtain the optimal phase shift $\Phi_{2,n}^{\varepsilon +1}$ based on Equation (Equation 15);       Given $\left\{\Phi_{1,n}^{\varepsilon +1},\Phi_{2,n}^{\varepsilon +1},q_{2,n}^{\varepsilon }\right\}$, obtain the optimal trajectory $q_{1,n}^{\varepsilon +1}$ based on Equation (Equation 23);       Given $\left\{\Phi_{1,n}^{\varepsilon +1},\Phi_{2,n}^{\varepsilon +1},q_{1,n}^{\varepsilon +1}\right\}$, obtain the optimal trajectory $q_{2,n}^{\varepsilon +1}$ based on Equation (Equation 18);       Update the iterative number ε=ε+1;**until** convergence.

The sum rate is upper-bounded due to the constraint of *P*. Moreover, as shown in the alternating optimization steps of Algorithm 1, each iteration consists of two subproblems, both of which are solved to their respective optimal values. The iterative process of the algorithm satisfies the property of monotonic non-decrease. Based on the Monotone Bounded Convergence Theorem, it can be concluded that the sum rate converges to a local optimal solution.

The computational complexity of the joint IRS phase shift and UAV trajectory optimization algorithm in this paper is dominated by its iterative decomposition framework and core input scales. The algorithm converges after a constant number of outer iterations $L_{\mathrm{outer}}$, with each iteration involving two sub-problems. For IRS phase shift optimization, since the optimal phase of each of the $m_{1}/m_{2}$ IRS elements is computed via 𝒪(1)arg(·) operations and phase adjustment is required for each of the *N* UAV trajectory discretization slots (core input scale), its complexity per outer iteration is $𝒪(N\cdot \left(m_{1}+m_{2}\right))=𝒪$. For UAV trajectory optimization, the non-convex problem is transformed into convex sub-problems via SCA. Each convex subproblem, solved by interior-point methods, involves 𝒪(N) variables and 𝒪(N) constraints, leading to a complexity of $𝒪(N^{3})$. Summing the two subproblems and incorporating $L_{\mathrm{outer}}$, the overall time complexity of the algorithm is $𝒪\left(L_{\mathrm{outer}}\cdot \left(N+N^{3}\right)\right)=𝒪\left(N^{3}\right)$.

Figure 2 illustrates the logic and process of this work carried out.

## 5. Numerical Results

This section conducts simulation experiments to demonstrate the effectiveness of the joint trajectory optimization algorithm. Establish a three-dimensional Cartesian coordinate system with the BS as the origin. The closer coverage holes have four sensor nodes, while the farther ones have three nodes, namely, $K_{1}=4$, $K_{2}=3$. BS is located at (0,0,0), and $u_{1}=\left(15,-20,0\right)$, $u_{2}=\left(15,50,0\right)$, $u_{3}=\left(30,70,0\right)$, $u_{4}=\left(40,30,0\right)$, $s_{1}=\left(15,160,0\right)$, $s_{2}=\left(25,200,0\right)$ and $s_{3}=\left(40,140,0\right)$. We assume that the communicating order in one working cycle is predetermined as $u_{1}\to u_{2}\to u_{3}\to u_{4}\left(\mathrm{and}s_{1}\to s_{2}\to s_{3}\right)$. The UAVs fly at a fixed altitude H=100 m with the maximum flight speed $V_{1max}=5$ m/s and $V_{2max}=5$ m/s. The spacing of the IRS elements is d=λλ22, where λ=0.05 m. Other parameters are set as $\beta_{0}=-30$ dB, P=50 dBm, $\sigma^{2}=-90$ dBm, $m_{1}=150$, $m_{2}=150$ and $R_{l\mathrm{ow}}=0.01$ Hz/bps [16].

Figure 3 shows the flight trajectories of each period within one working cycle with T=60 s. We use different colors to distinguish the current communication nodes involved. The UAV equipped with the IRS changes its trajectory to approach the nodes as the mission period increases.

To further demonstrate the effectiveness of the proposed algorithm, three distinct trajectory optimization schemes are introduced for comparison. It is well established that the maximum signal-to-noise ratio (SNR) is achieved when the IRS is positioned close to the nodes or BS, whereas an active relay is typically deployed between the node and BS [2]. Therefore, two traditional trajectory schemes are introduced as benchmark schemes in this paper: one is that the IRS-UAVs fly around nodes along a straight-line trajectory; the other is that the IRS-UAVs hover at fixed points within each coverage hole, respectively. (1) Proposed scheme, which is given by Algorithm 1. (2) Line1 scheme, where the UAV flies from one node to the next at constant speed. (3) Hovering scheme, where the UAVs hover in the geometric center. The times of different communication pairs are equally distributed in the comparison scheme.

Compared to the traditional trajectory schemes, it can be observed that the flight trajectory of the UAV within one cycle exhibits a similar trend to the inter-node connection line. Nevertheless, due to multiple constraints such as the UAV’s maximum flight speed, working cycle and so on, the circular trajectory is always shorter than the inter-node connection line.

Meanwhile, as this paper assumes the communication link to be a LoS link, Node $s_{k2}$ in the farther coverage hole often achieves a higher transmission rate than Node $u_{k1}$ in the closer coverage hole, as it constructs an uplink via both dual-reflection and single-reflection links. This also explains why the flight trajectory of $I_{2}$ is more consistent with the traditional trajectory than that of $I_{1}$.

Figure 4 illustrates the comparison between the proposed optimization scheme and the traditional scheme in terms of each period and sum rate. Under the traditional trajectory, nodes cannot ensure that the transmission rate remains optimized throughout the transmission time; thus, the proposed algorithm is significantly superior to the unoptimized traditional scheme. Furthermore, as the optimized flight trajectory is shorter than the traditional one, the UAV may hover at a certain position until the next node requires service.

Figure 5 shows the flight trajectories of UAVs during one working cycle with T=60 s, 36 s, and 24 s, respectively. As the time length of working cycles increases, the UAV has more time to fly towards the theoretically optimal transmission position. Near the optimal positions, UAVs may be in a hovering state.

Figure 6 and Figure 7 show that the UAV tends to fly closer to the node at the fastest speed. As the working cycle increases, the UAV has more time to fly at low speed or hover near the node to assist in communication with the same node and finally fly to the next node. When the speed drops to 0, it means that the IRS-mounted UAV is hovering to serve the same node. This can also be derived from the fact that each of the hovering points has the same color as its neighboring points in Figure 5.

As shown in Figure 8, given the differences in QoS constraints of wireless sensor nodes across various communication scenarios, this paper further investigates the flight trajectories of UAVs under different QoS constraints. It is shown that as the stringency of QoS constraints increases, the flight trajectory of UAV1 gradually approaches the BS, while the flight trajectory of UAV2 remains relatively unchanged.

Table 1 presents the variation in optimization rates under different total numbers of reflecting elements. It can be observed that when the number of elements is small, increasing the number of elements leads to a significant improvement in the optimization rate; however, as the number of elements continues to increase, the magnitude of the optimization rate growth gradually diminishes. Meanwhile, to maximize the optimization rate, the deployment logic of reflecting elements should adhere to the following principle: long-distance communication, which has a higher demand for IRS gain, requires a larger number of elements, and node-dense areas need more elements to cover additional communication links.

## 6. Conclusions

This paper proposes a novel double IRS-UAV-assisted framework for the uplink of WSNs to enhance the pairwise sum rate. Unlike existing studies that mainly focus on single-reflection links or ignore IRS interconnections, this work enables the BS to serve proximal nodes via a single-reflection link of one IRS-UAV and distal nodes through a double-reflection link of two IRS-UAVs. A joint optimization algorithm is designed to maximize the sum rate by co-optimizing UAV trajectories and IRS phase shifts, with constraints on minimum QoS for each node. The original non-convex problem is decomposed into phase shift optimization and trajectory optimization sub-problems, solved iteratively using techniques like SCA and slack variables. Numerical results validate that compared with the traditional trajectory scheme, our proposed dual IRS-UAV collaborative optimization algorithm improves the system sum rate by 11.91%; compared with the hovering trajectory scheme, the sum rate is increased by 16.36%.

When the number of WSN nodes increases significantly (i.e., when there are nodes that fail to meet the QoS constraints under the current dual IRS-UAV configuration), an additional set of IRS-UAVs can be considered. Specifically, we will reallocate these nodes that do not meet QoS requirements to the newly added IRS-UAV or the original IRS-UAV based on the distance between nodes and IRS-UAV, as well as the signal strength of communication links, to ensure all nodes satisfy the QoS constraints.

## Figures and Tables

**Figure 1 sensors-25-06265-f001:**
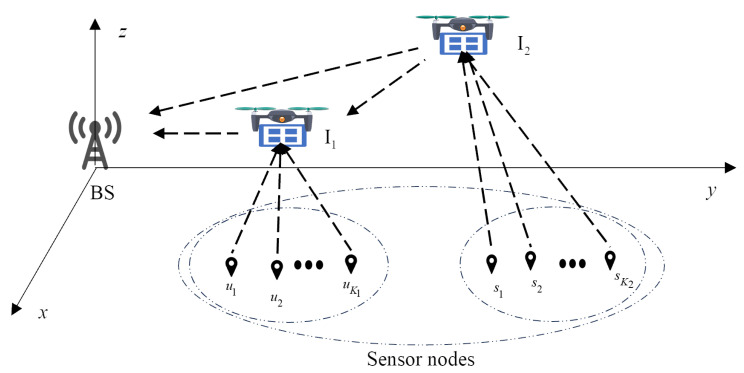
Double IRS-UAV-assisted uplink of WSNs.

**Figure 2 sensors-25-06265-f002:**
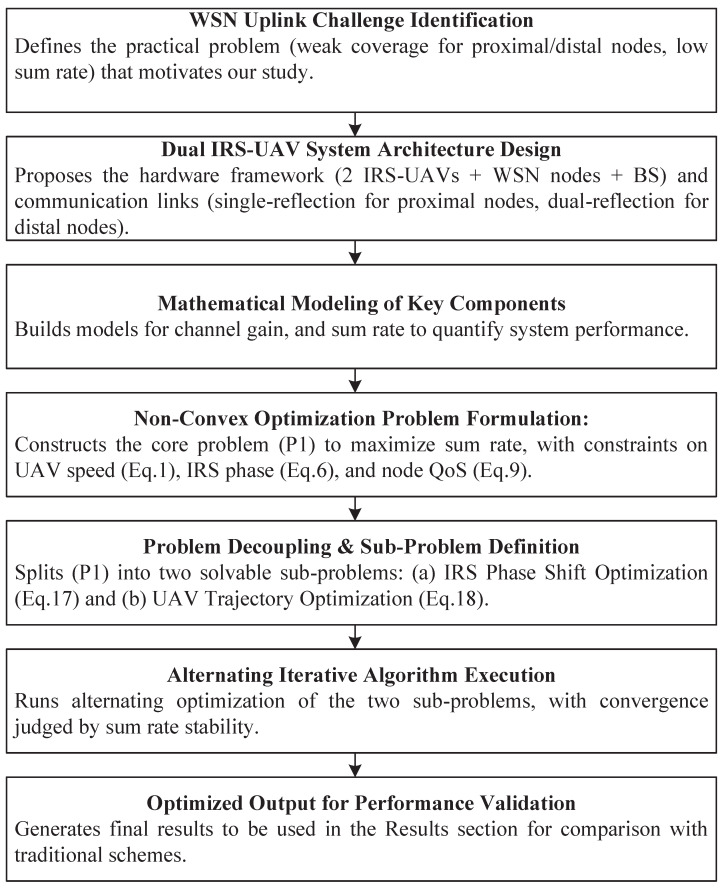
Workflow chart.

**Figure 3 sensors-25-06265-f003:**
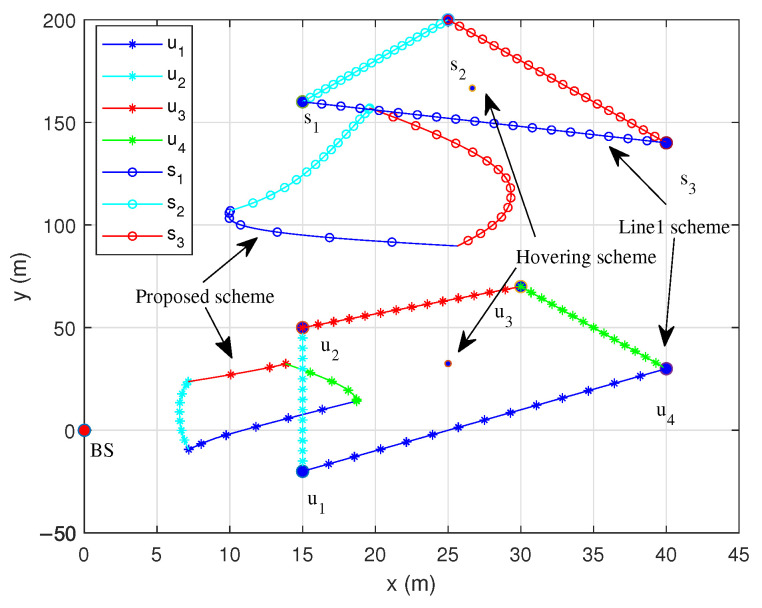
Trajectories of different schemes.

**Figure 4 sensors-25-06265-f004:**
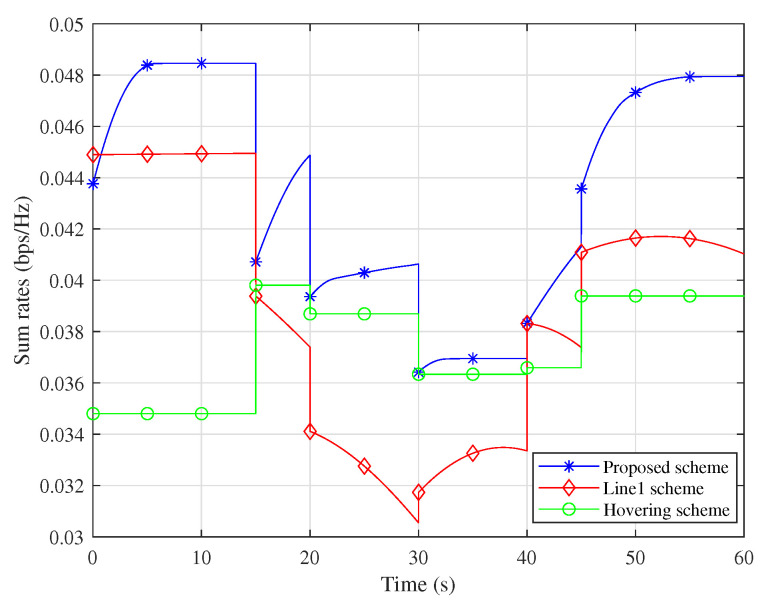
Comparison of sum rates with different scheme.

**Figure 5 sensors-25-06265-f005:**
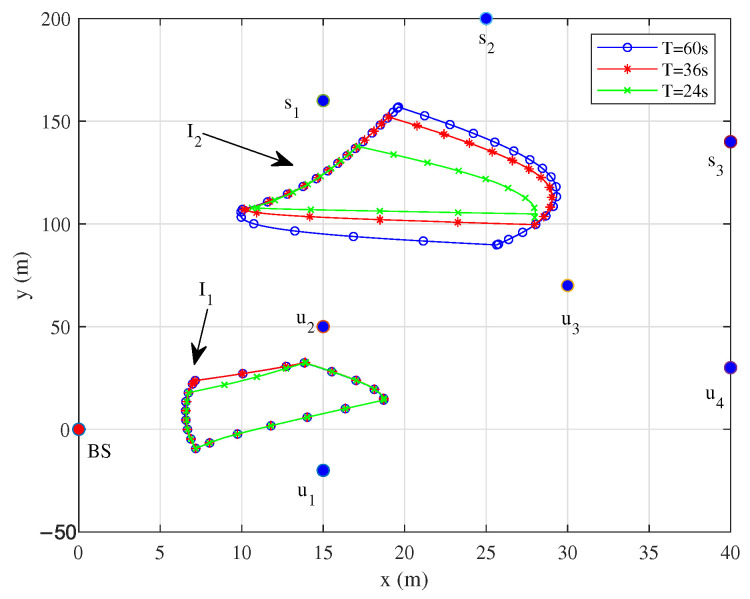
Trajectories under different *T*.

**Figure 6 sensors-25-06265-f006:**
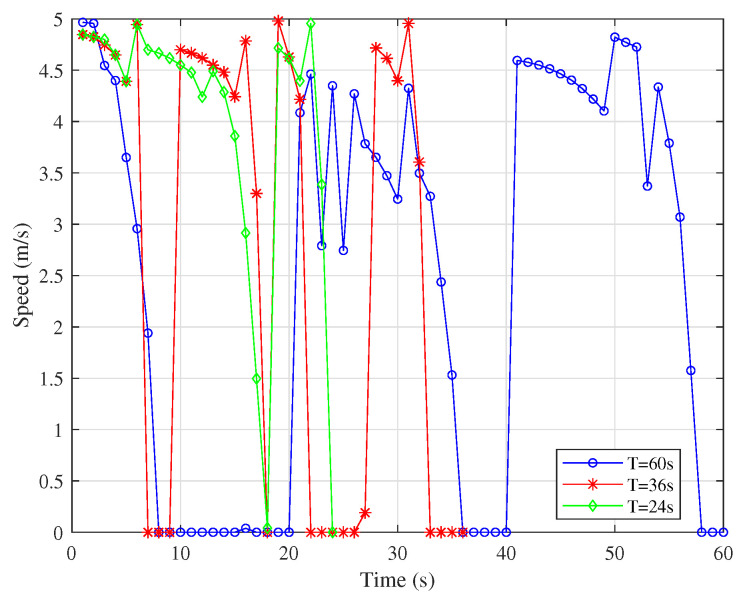
Speeds of UAV1 under different *T*.

**Figure 7 sensors-25-06265-f007:**
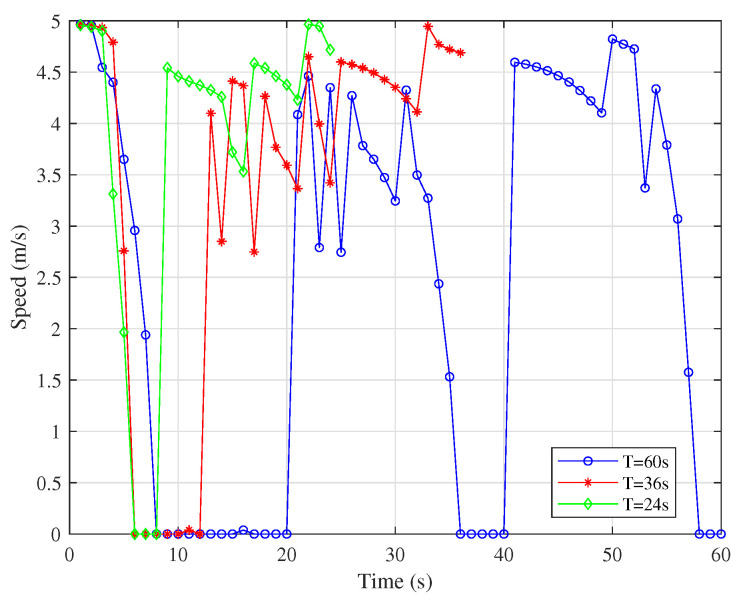
Speeds of UAV2 under different *T*.

**Figure 8 sensors-25-06265-f008:**
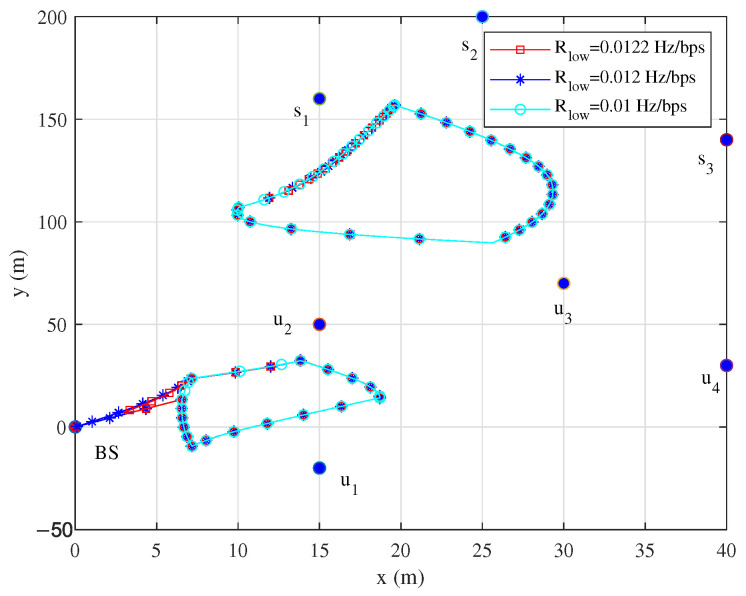
Trajectories under different $R_{\mathrm{low}}$.

**Table 1 sensors-25-06265-t001:** Optimization rates under different numbers of reflecting elements.

$m_{1}+m_{2}$	Compared to Line 1 Scheme	Compared to Hovering Scheme
300	11.91%	16.36%
400	13.56%	17.65%
500	14.11%	18.35%
600	14.63%	18.77%
1000	14.94%	19.65%

## Data Availability

Dataset available on request from the authors.

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
