# Peer review of "Joint Trajectory and IRS Phase Shift Optimization for Dual IRS-UAV-Assisted Uplink Data Collection in Wireless Sensor Networks"

_sensors, 2025, doi:10.3390/s25206265_

Round 1

Reviewer 1 Report

Comments and Suggestions for Authors

Dear Authors,

Thank you for your work and interesting paper. Please find below my comments

1- the title needs to be revisited. although descriptive and informative, it contains abbreviations, which would be better to avoid. Either with the complete word or rephrase the title.

2- Abstract: is descriptive, highlight the technical task and the achieved results. I would be better to highlight a bit on used method without the need to proof. i.e. using the word "algorithm" is extremely generic and does not give indication about the nature of your solution. this comment can be taken as recommendation.

3- Using the word UAV is again very generic. where are you really trying to improve the result according to the QoS. are the miniature UAV vulnerable, MALE UAVs? what are the tasks that can benefit from improving/ resolving the aforementioned technical task? coverage, connectivity? is it going at all to be beneficial for GPS/GLONASS less flight missions? framing the technical task is quite important here

4- Equations: well defined. variables are also explained.

5- you can push your paper further more by highlighting the effect of IRS on the UAV performance especially in low batter mode. as it reduces the power needed for communication. it is also very usable for smart cities applications and disaster recovery. 

6- Reflecting back on the type of the UAV, as you know UAVs are mobile, and IRSs need to be dynamically configured to reflect signals optimally. Coordinating both in real time is a complex optimization problem. so for different UAV types, the problem will be represented in different level of challenge. and that will lead us to the following comment

7-  since we have different type of UAV which are battery/energy dependent, it is important to understand that IRS (though passive) still require control circuitry. Balancing communication performance with energy consumption is a multi-objective optimization problem.

8- Now for fast moving UAVs, IRS configurations must adapt instantly to changing environments and since IRS have typically have slow reconfiguration speeds, it is important to state some constraints here.

9- What are the recommendations of the authors regarding the scalability and deployment of their system? in terms of setup for the IRS, I mean. would that work on certain altitude, say the IRS is installed on four of high rise buildings/ tower? what are the precautions to be taken into consideration.

10-  simulation results. adequate, based on the proposed algorithm, but the trajectoiries are extremely smooth for such hard technical task. this is due to the constraints and simplification of task (which is good, you knew how to frame the work, in fact, this is an extremely challenging technical task)

Reviewer 2 Report

Comments and Suggestions for Authors

This paper considers an IRS-UAV system composed of two UAVs that reflect the uolink signal to communicate with nodes in a short range and the BS placed further away. This is done by optimizing the flight trajectories of UAVs and the beamforming of the IRS.

Given the trajectories of the UAVs and the phase shifts of the reflected signals, the authors aim to maximize the QoS of the system. This is not an easy problem to solve. As such, the authors propose an algorithm to maximize the sum rate. 

The paper is well written and organized. Their proposed algorithm effectively maximizes the QoS considering the UAVs' trajectories and phase shifts. 

I have the following concerns and observations:

1-The authors consider that the sensor nodes’ location is known. In many cases, it is not possible to know the exact location of nodes, and during the system operation, nodes can be displaced and change their location. Authors should explain in detail how their proposed schemes can work with random positions of nodes.

2-Also, 7 nodes in a WSN is a rather low number of nodes in such systems. How can these schemes be used for WSNs composed of hundreds of nodes?

3-The sentence “received signals to users” is very confusing. Does it receive it from users? Or does it receive it to then transmit it? 

Reviewer 3 Report

Comments and Suggestions for Authors

First of all thank you for providing me a chance to review your paper. I have found some flaws in your paper. Those flaws are provided below in the form of pointers.

  1. The title of you work is a bit confusing, paraphrase it in a way so that the meaning can be understood clearly.
  2. Use WSNs instead of WSN in the whole manuscript.
  3. In the last line of your abstract “Numerical results indicate that this algorithm can effectively optimize flight trajectories of the UAVs and significantly enhance the pairwise sum rate of the system.” Please mention the achieved optimization rate in percentage too.
  4. There is no need to provide abbreviation in the keywords section, only the full for is enough to be presented.
  5. The first paragraph of the introduction section from line 38 to 43 can be considered as an introduction but rest of the introduction section is basically the related works. So please re write a new introduction section that can make a base for the readers to understand the rest of the technical achievements in the manuscript easily that you have presented.
  6. At the end of the introduction you can keep the last paragraph of the introduction section now in present form of the article as the key achievements of your study.
  7. Why has a related works section not been included after the introduction? I strongly recommend adding one, as it is an essential part of a manuscript.
  8. At the end of the related works section, include a paragraph outlining the key factors and insights that motivated you to undertake this study.
  9. Bring figure 1 before the section 2 so the readers can refer to it while reading the manuscript.
  10. In equation 1 you have given (m,n), you have defined what n is about but there is no information on m, secondly the structure of the equation is not understandable, like what is (1,2) on the right hand side on the equation, correct it if I am write, and if I am wrong then explain it what does this structure of the equation means.
  11. Have a look into equation 11 there is no right or left hand side of the equation mentioned and again I am seeing (1)(6)(9), can you please explain what kind of a structure of the equation is this you are following.
  12. Same problem with equation (17) and (18)
  13. The caption of the table 1, correct it.
  14. In section 2 provide a table that should include all the parameters of the scenario you have created as a WSNs, to achieve the desired output.
  15. Before the results section provide a flow diagram that should elaborate a block wise explanation of you work
  16. Before the results section provide a pseudo code of your work, so that if some readers want to recreate this work for their own understanding than they may easily can do so.
  17. From your results the readers may not get proof of convergence to at least a local optimum, because it’s a known fact that if you donot provide a convergence guarantee in a WSNs, the algorithm practicality is considered uncertain.
  18. You have not provided any results regarding the scalability of the opted methodology either in tabular or line graphs which ever suits you better.
  19. And in you work I have noted that you have only provided information on the Line trajectory and Hovering trajectory, you should must provide some comparison results against other learning based approaches, for the better elaboration of your work.
  20. The conclusion can be written in a much better way where you can include the achieved optimization results of you opted algorithm in the form of percentage.
  21. In the results section include a table that encapsulate your achieved results in the form of percentage.
Comments on the Quality of English Language

Must be improved

Round 2

Reviewer 1 Report

Comments and Suggestions for Authors

Thank you for your revised work

Author Response

Dear Reviewer, ​ We are truly grateful for your recognition of our research work in the review. Your positive feedback on the novelty of the dual UAV-IRS framework and the rigor of our algorithm design means a great deal to us, as it affirms the value of our efforts.​   We will continue to polish the paper with care to better present our work. Thank you again for your valuable time and encouraging comments.​   Sincerely,​ Heng ZOU, Hui GUO

Reviewer 3 Report

Comments and Suggestions for Authors
  1. The introduction of the paper is still not appropriate; rewrite it and make it lengthy, up to almost between 1000 to 1500 words.
  2. Make the related works a separate section as a main heading, not a subheading of the introduction.

For the rest of the changes, I agree with them.

Author Response

Dear Reviewer, ​ Thank you sincerely for your constructive feedback on our paper’s introduction. We fully acknowledge your point that the introduction needed further refinement to be more comprehensive and logically robust. Following your guidance, we have thoroughly revised and expanded the introduction and made the related works a separate section as a main heading. ​ The revisions focus on three key areas to better contextualize our work and lay groundwork for subsequent sections. First, we added detailed explanations of IRS’s technical traits: we clarified its composition , core advantages, and how it outperforms traditional active components  in mitigating coverage gaps. Second, we elaborated on IRS’s real-world applications—covering smart cities, industrial IoT, and environmental monitoring—to highlight its practical value. Third, we detailed the rationale for UAV-IRS integration: we analyzed fixed IRS’s limitations in dynamic scenarios (e.g., moving users) and explained how UAVs’ mobility complements IRS, citing prior studies to validate this combination while noting gaps our work addresses. ​ These additions strengthen the introduction’s logical flow, linking technical background to our proposed dual UAV-IRS framework and making the transition to subsequent sections  smoother. ​ We greatly appreciate your input and would welcome any further suggestions to enhance the paper. ​ Sincerely,​ Heng ZOU, Hui GUO